


# Lightning and thunder explanations in encyclopedias - from ancient Greece to WIKIPEDIA

Kristian Schlegel, Wolfenbüttel, Germany

*Correspondence to*: Kristian Schlegel (kristian.schlegel@copernicus.org)

**Abstract.** After a brief introduction to encyclopedias, the explanation of lightning and thunder is examined in well-known encyclopedias from Greek philosophers to modern physics. Starting with Aristotle (who is not regarded as encyclopedist, but very important for our topic), ten out of more than two hundred known encyclopedias are treated in some detail. This selection is certainly somewhat arbitrary, but it was attempted to choose encyclopedias which are highlights and were widely circulated at their time. In antiquity and during the Middle Ages the explanations of thunderstorms were generally wrong, for

instance explaining lightning as a consequence of thunder. Besides, strange and often weird effects of lightning were reported. Many authors of those times used explanations of former encyclopedias sometime referring to earlier authors, often just plagiarizing. These wrong and strange ideas persisted for almost two millennia in encyclopedias. From the middle of the 18th century onward physical explanations begin to emerge which are still valid today. More and more correct details of lightning and thunder and the results of experiments are reported in encyclopedias. It is also attempted in this manuscript to

name insights of other scientists which the authors of contemporary encyclopedias do not mention, but which should have been available at that times. Finally, it is stated that even today not all details of thunderstorms are well understood.

## 1 Introduction: Encyclopedias

The term encyclopedia was composed of the two words of the Greek phrase "enkyklios paidia" meaning „comprehensive" and „for the education of youth". It was used for „the circle of arts and sciences, the course of study which every Grecian

youth went through before entering upon professional studies" (Liddell, et al., 1940). Ancient and medieval authors did not use this term, they named their work with terms like „Naturalis Historia" or fanciful names like „Hortus deliciarum" (see below). The German scholar Johannes Aventinus (1477-1534) was the first author who used the term encyclopedia in 1517. Encyclopdias are supposed to contain the actual knowledge of the respective age which has been accepted by the authorities of the time, relaying often to earlier scholars. Their widespread circulation indicates that the readers appreciated exactly this

purpose.
A very comprehensive work about encyclopedias (,,enc." in the following) is the publication of Collison (1966). He treats more than 230 enc., about 50 in greater detail, displaying their tables of contents and discussing different editions. The book

covers the time from ancient Greek to the middle of the 20th century on 334 pages. The entry „List of encyclopedias by date" in the English-language Wikipedia contains about the same number of enc.

In Roman times and in the Middle Ages the authors tried in general to summarize the entire knowledge of the time in their enc.. They included usually theological topics (God, angles, saints), but also entries about humans, medicine, animals, plants, minerals, a popular entry was „about monsters". An important part were also the „seven liberal arts", i.e. rethoric, grammar, logic (trivium), and astronomy, arithmetic, geometry, and music (quadrivium). These subjects, already mentioned by CiceroA (106-43 B.C.) and Seneca (ca. 1-65), were also for a long time taught at universities.

Beginning in the 18th century enc. of special topics were published, like for theology, history, poetry, music and technical subjects, as well as for countries and states. In antiquity and in the Middle Ages enc. were generally published by one single author. Starting in the 17th century quite often several authors contributed to an enc. (see below).

The purpose of this manuscript is not a review of lightning research, but an attempt to follow the ideas and causes about thunderstorm phenomena through the past centuries in well-known enc. The selection of enc. treated in the following may be

somewhat arbitrary, but it was attempted to choose enc. which were some kind of highlight ore milestone in the history of enc.

A concise history of lightning matters beyond enc. was published by Bouquegneau (2011), a thorough treatment of the physics of lightning by Dwyer and Uman (2014).

**2. Ancient Greek Philosophers**

The Greek philosophers composed no enc., their wisdom and knowledge was handed down orally. Only later, scholars wrote it down. **Aristotle** (384-322 B.C.) is therefore not regarded as an encyclopedist, but since his influence on subsequent scholars was eminent, his statements about lightning and thunder should briefly be discussed. In his METEOROLOGICA

(Book II, Part 9, Aristotle in The Internet Classics Archive, 2024) he explained:

> *As we have said, there are two kinds of exhalation, moist and dry, and the atmosphere contains them both potentially. It, as we have said before, condenses into cloud, and the density of the clouds is highest at their upper limit…. Now the heat that escapes disperses to the up region. But if any of the dry exhalation is caught in the process as the air cools, it is squeezed out as the clouds contract, and collides*

> *in its rapid course with the neighbouring clouds, and the sound of this collision is what we call **thunder**… It usually happens that the exhalation that is ejected is inflamed and burns with a thin and faint fire: this is what we call **lightning**, where we see as it were the exhalation coloured in the act of its ejection. It comes into existence after the collision and the **thunder**, though we see it earlier because sight is quicker than hearing.*

Here the most important (erroneous) assumption is that thunder precedes lightning. Aristotle admits however, that there are other views: **Anaxagoras** (ca 500-427 B.C.) stated that the upper region of the atmosphere is composed of a flaming hot substance (ether) which descends from above. Lightning is then the gleam of this fire, and thunder the hissing noise of its extinction in the cloud. **Empedocles** (ca. 455 B.C.) had similar ideas. Aristotle does not believe this, he argues:

> *But this involves the view that **lightning** actually is prior to **thunder** and does not merely appear to be*
> *so. Again, this intercepting of the fire is impossible on either theory, but especially it is said to be drawn*
> *down from the upper ether. Some reason ought to be given why that which naturally ascends should*
> *descend, and why it should not always do so, but only when it is cloudy.*

Aristotle's view of lightning and thunder prevails for the following two millennia. This means that most of the enc. treated in the following accept his view, sometimes with reference to him sometimes just plagiarized. It is nevertheless remarkable to
explore the additional views and comments written in various enc.

**3. Roman Encyclopedias**

**Marcus Terentius Varro** (116 -27 B.C.) was a polyhistor who contributed significantly to the development of enc.
Unfortunately his work DISCIPLINARUM LIBRI IX which contained the content of what was later called „the seven liberal arts" is lost  and only fragments have survived (Simon, 1964). Therefore we do not know if he wrote about our subject.

The NATURALIS HISTORIA (published 77 AD) by **Pliny the Elder** (23-79) is truly the first real enc. and became an editorial model for future enc. (Fig. 1). It is composed of 37 books (for citations in this paper: the English Translation of the
Internet Archive, 1967), Book 2 treats Meteorology . Pliny follows Aristotle in explaining the cause of lightning and thunder, he gives proper credit to him (In Book 1, entitled „List of content and authorities" Aristotle is named many times). In Sections XLII and XLIII (Book 2) Pliny clearly states that lightning is a consequence of thunder, and also mentions (Section LV) that

> *…the flash is seen before the thunderclap is heard (this not being surprising, as light travels more*
> *swiftly than sound); but that Nature so regulates the stroke of a thunderbolt and the sound of the thunder*
> *that they occur together…*

There is however a new statement (section XXXVII), not existent in Aristotle's Meteorologica:

> *I have seen a radiance of star-like appearance clinging to the javelins of soldiers on sentry duty at night*
> *in front of the rampart; and on a voyage stars alight on the yards and other parts of the ship, with a*
> *sound resembling a voice, hopping from perch to perch in the manner of birds.*





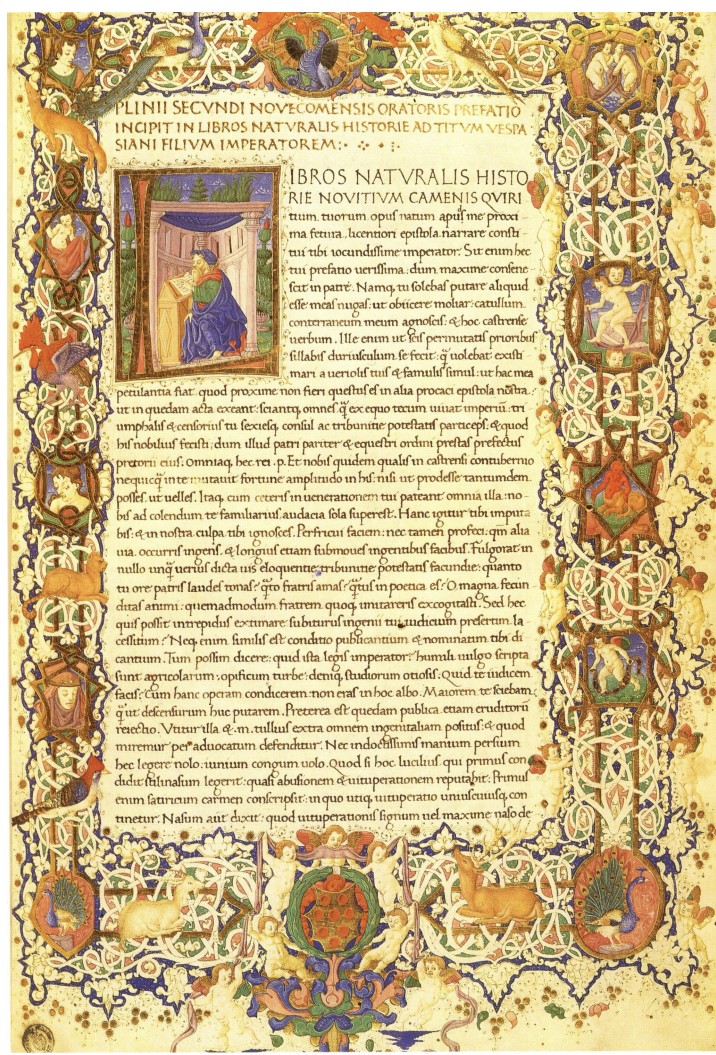

Fig. 1: Pliny the Elder, Natural History in ms. Florence, Biblioteca Medicea Laurenziana, Plut. 82.4, fol. 3r., Public domain, via Wikimedia Commons.

This is most likely the observation of what we call St. Elmo's fire today.

The „bright thunderbolts" mentioned in section LII could be a ball lightning, reported today to move through walls:

> *Of thunderbolts themselves several varieties are reported. Those that come with a dry flash do not cause a fire but an explosion. The smoky ones do not burn but blacken. There is a third sort, called 'bright thunderbolts,' of an extremely remarkable nature; this kind drains casks dry without damaging their lids and without leaving any other trace, and melts gold and copper and silver in their bags without singeing the bags themselves at all, and even*
*without melting the wax seal.*

Some other statements sound rather bizarre, like (section LV):



*Lightning unaccompanied by thunder occurs more often by night than in the daytime. Man is the one creature that is not always killed when struck—all others are killed on the spot; nature doubtless bestows this honour on man because so many animals surpass him in strength. All things (when struck) fall in the opposite direction to the flash.*
*A man does not die unless the force of the blow turns him right round. Men struck from above collapse. A man struck while awake is found with his eyes shut; while asleep, with them open.*

There are more remarkable entries in other books of Pliny's work, like in Book 15 (The natural history of fruit trees), section XL:

*...the laurel alone of all the shrubs planted by man and received into our houses is never struck by lightning.... It is*
*stated that the emperor Tiberius used to put a wreath from this tree on his head when there was a thunder-storm as a protection against danger from lightning.*

In Book 18 (covers Agriculture and Meteorology), section LXXXI he attempts some kind of weather forecast:

*A thunderstorm in summer with more violent thunder than lightning foretells wind in that quarter, but one with less thunder than lightning is a sign of rain. If there are flickers of lightning and claps of thunder in a clear sky, there*
*will be stormy weather, but this will be extremely severe when it lightens from all four quarters of the sky; lightning in the north-east only will portend rain for the next day, and lightning in the north a north wind. Lightning on a fine night in the south, west or north-west will indicate wind and rain from the same quarters. Thunder in the morning signifies wind, and thunder at midday rain.*

Finally a quotation from Book 36 (Treats of gemstones and other precious stones), section LV:
*'Brontea,' or 'thunder stone,' which is like the head of a tortoise, is supposed to fall from thunderclaps and to extinguish fires where lightning has struck, or so we are led to believe.*

There are other roman enc., but they do not contain entries about our subjects or just repeat Pliny, like **Gaius Julius Solinus**' COLLECTANEA RERUM MEMORABILIUM (3rd Cent).

**4 Middle Ages and early modern times**

According to Collison (1966, p 33f) the bishop **Isidore of Seville** (560-636) compiled the first „Christian" enc., the ETYMOLOGIAE (published around 623, English translation from Latin: Barney, 2006). It contained 20 books and covers many theological items, but also medicine, law, languages, agriculture and wars. In Book 13 there are two entries which are
interesting here: „About Thunder" and „About lightning". In principle they are a repetition of Aristotle's explanations, but contain an interesting sentence, listing three types of lightning for which three different latin words are used „fulgus" which touches, „fulgur" which incites and burns, and „fulmen" which cracks (In English translations: thunderbolts). The latter two expressions are frequently used in all latin enc. when the different effects of lightning are explained.

- - -




The first enc. composed by a woman was the HORTUS DELICIARUM (Garden of Delights, published 1175) from **Herrad of Landsberg**.(about 1128-1195). She was abbess of the Hohenburg Abbey in Alsace. Her exceptional work, written in Latin on 648 pages, was illustrated by 336 illustrations (see Fig. 2, as an example). Unfortunately the original was destroyed 1870 in Strasbourg during the Franco-German war, but the work could be reproduced from earlier copies. Under the section of the liberal arts, chapters 73 and 75   deal with thunder and lightning (Hildebrand, 1616). Herrad follows in principle

Aristotle's explanation, but in quite flowery terms. She states that the lightning is not red like fire, but of a „subtil" matter which is white or yellow. It is called „fulgur" and is not dangerous, contrary to „fulmen" which destroys. (see above). She compares the sound of thunder with an explosion of a mountain or fortress by gun powder. Dark and thick clouds cause a louder uproar than thin ones. Sheet lightning without thunder happens when a cloud is thin, so that the fire can escape easily without a bang. Other strange statements are:

*A thunderbolt can kill an embryo without harming the mother. If a thunderbolt hits a barrel of wine, the barrel may remain intact, but the wine is poisoned and men or animals drinking it will die.*

It should be noted that Herrad mentioned the famous Benedictine abbess and polymath **Hildegard von Bingen** (1098-1179) several times as a friend and teacher.

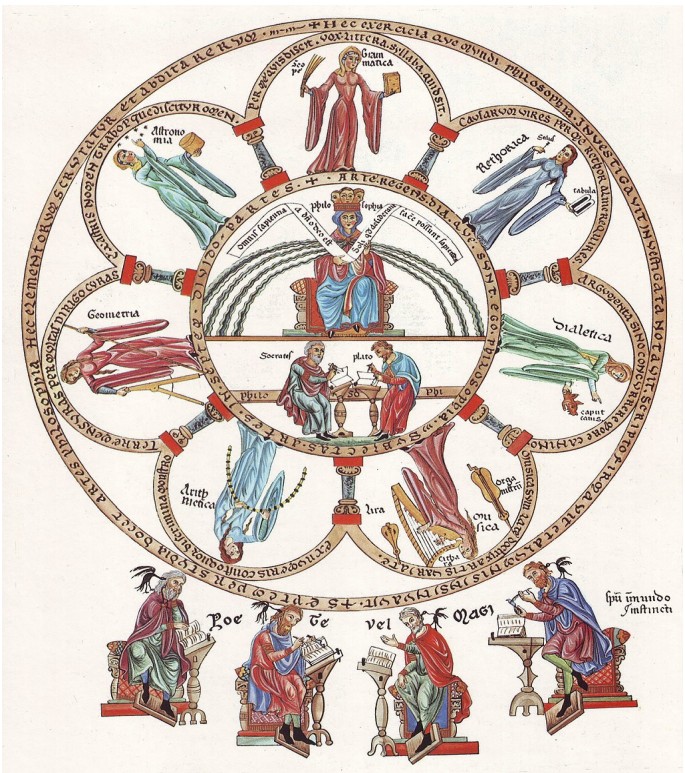

**Fig. 2**: Allegorisation of the philosophy (above Socrates and Plato) surrounded by the seven liberal arts, illustration from the

Hortus Deliciarum (Dnalor-01, Wikimedia Commons License CC-BY-SA 3.0, high resolution file is available)





Worth to mention is LIBER DE NATURA RERUM (published between 1225 and 1241) composed by the Flemish Dominican friar **Thomas Cantimpratensis** (1201-1272). The work, structured in 20 books, was largely widespread in the Latin Late Middle Age. Book XIV „On air motion" contains details of our topic (Cantimpré, 2022). He gives credit to

Aristotle and repeats his ideas about thunder and lightning. Remarkably, his work was translated into German already in 1349 by **Konrad von Megenberg** (1309-1374), a cathedral cannon in Regensburg. His work DAS BUCH DER NATUR (Pfeiffer, 1994) was called „the first  natural history in German language" and was widely circulated in German speaking countries. It is not a word-by-word translation of Thomas' work, but Konrad took the liberty to rearrange the content and left out certain parts. Within his book II „Von den Himmeln und von den siben Planeten" (about heaven and the seven planets)

chapter 25 deals with „Von dem donr und von den plitzen" (on thunder and lightning). Its content agrees well with Thomas' chapter „de Tonitruo" (about thunder, Book XIV,1). This subsumption proves again that the topic meteorology was included in astronomy at that times. Konrad repeats many of the impacts of thunder and lightning already mentioned. An interesting one is

> *„Some people think the thunder is a stone which falls down during thunderstorms. This is not true, since if the thunder is a stone it would cause wounds at men and animals which are hit. People which are hit have no wounds, but are black, since they are burned…*

The possible sign of a ball lightning, already mentioned in Pliny's work is repeated, as well as the protective power of laurel. It is also stated that lightning harms  the blossoms on trees.

----

There are several other well-known medieval enc., like for instance FONS MEMORABILIUM UNIVERSI by **Domenico Bandini** (1335-1418), or MARGARITA PHILOSOPHICA by **Gregor Reisch** (1470-1525), but they do not contain any new information about our subject. This is also true for the SYLVA SYLVARUM which was composed by **William Rawley** (1588–1667) from the material **Francis Bacon** (1561-1626) had collected for his planned voluminous enc. THE PHENOMENA OF THE UNIVERSE which however was never published.

175 ----

Collison (1966) mentions in his book 12 enc. from the Arabic and Persian culture area. As an example the ULTIMATE AMBITION IN THE ARTS OF ERUDITION of the Egyptian scholar **al-Nuwayri (**1279-1333) should be briefly mentioned here. It is a 9000-page work organized in five books. Book I ,entitled „Heaven and Earth", treats in Section 1.2 „Meteorological phenomena". Our topic is only briefly mentioned under the subheading „On shooting stars, thunderbolts,

thunder, lightning and rainbows":

> *As for thunderbolts, al-Zamakhshari wrote in his exegesis of the Quran: "A thunderbolt is a clap of thunder accompanied by a tongue of fire." They say that the bolt is struck when different parts of a cloud collide. It is a thin, sharp fire that catches onto anything ti strikes. Despite its intensity, it is quick to die out. However, when it strikes a palm tree, it burns the top of it.*

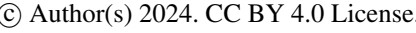

Thus the author uses Aristotle's view that thunder comes from colliding clouds, without referring to him. This is not remarkable, since it is well-known that Arabic scholars were familiar with the works of Greek philosophers. Following is a description of the possible ball lightning already mentioned in Pliny's work, that it burns things in a sack, without burning the sack itself. Further one can read:

> *If it [the thunderbolt] strikes the sea, it sinks into it and burns what is around it. Sometimes, when its fire is*
> *extinguished in the ground, it makes it cold and dry, and formations of rock or iron or brass are created from it,*
> *and sometimes the iron is used to manufacture swords that cannot be resisted.*

A remarkable religious explanation is given as well:

> *As for thunder and what has been said about it, God Most High said: "The thunder proclaims His praise."*
> *Q( 13:13) The exegetes say that the thunder is an angel in charge of the clouds and he has a steel cable that he uses*
> *to steer the clouds from one land to another, just like the camel driver steers his camel.*

A treatment of the enc. of the Asian (China, India and Japan) culture is beyond the scope of this article.

## 5. Enlightenment and later


**Denis Diderot's** (1713-1784) ENCYCLOPÉDIE has been published in several editions between (1751 and 1772, ENCCRE, 2024). Diderot was the main editor, but he worked together with **D'Alembert** (1717-1783) and from 1760 on with **Louis de Jauncourt** (1704-1779). Our topic is treated under three entries: lightning (foudre, Vol VII, p 213, eclair, Vol V, p268)), and thunder (tonnerre, Vol 16, p412). Typical for the enlightenment is that from now on experiments and observations are cited
and not only opinions and beliefs.

Diderot's enc. is the first where a possible relationship to electricity is mentioned:

> *The matter of lightning appears to be the same as that of electricity.*

Reference is made to **Pieter van Musschenbroek** (1692-1761, inventor of the „Leyden-jar") and his lightning observations in Utrecht, and about global occurrence of lightning.
In the entry „eclair" more properties of lightning are summarized:

> *...the flame travels from one end to the other with great speed ... everything bursts and disperses with astonishing*
> *violence, and we then hear this noise which resounds in the air, and to which we give the name thunder, and of*
> *which lightning is the harbinger.*

This is an important progress, Diderot and his fellow editors do apparently not share the so far prevailing view that lightning
is the cause of thunder. He also gives for the first time a recipe to estimate the distance of the lightning, by counting the seconds between lightning and thunder, the distance is then 173 Toise (339 m) per second.

Under the entry „tonnerre" we read:

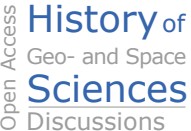

*Thunder is generally assumed to be caused by colliding clouds. A more satisfactory solution to the question has since been given, namely that thunder is not caused by clouds falling on top of each other, but by fire which*

*suddenly takes on sulfurous exhalations, and which causes noise when igniting.*

The smell is mentioned as well, although it is due to Ozone which was unknown at that time:

*...it is certain that the lightning is followed by a sulfurous vapor, as it appears by this taste of sulfur, which we feel after thunder and by this stifling heat which ordinarily precedes it.*

The rolling thunder is correctly explained:

*The clouds and objects that are on the surface of the earth reflect the sound, and multiply it almost like so many echoes.*

There are some remarkable statements which may refer to the fact that lightning causes nitric oxide which can react with ammonia in the air to the fertilizer ammonium nitrate:

*The utility of lightning is 1°. to refresh the atmosphere; in fact, we almost always observe that it is colder after it*

*has thundered: 2°. to purge the air of an infinity of harmful exhalations, and perhaps even to make them useful by attenuating them. It is claimed that the rain that falls when it thunders is better than any other to fertilize the land;*

The following remark refers to a commonly practice in earlier times: to ring church-bells at the approach of a thunderstorm (Bouquegneau, 2011), which is mentioned in several older enc.

*Lightning can be deflected by firing cannons; the sound of bells is a much less certain means; it sometimes*

*produces more harm than good…*

In many churches the corresponding bells showed the engraving „fulgura frango“ (I break the lightning stroke).

The ENCYCLOPÉDIE does not mention **Jean Antoine Nollet** (1700-1770) who became 1753 the first French professor for experimental physics. He performed various experiments with static electricity from 1740 on and was engaged with Franklin on opposite theories about lightning (Nollet, 1760)

-----

The first issue of the ENCYCLOPAEDIA BRITANNICA appeared 1771 in Edinburgh (Enc. Brit I, 1771). It was the idea of the bookseller **Colin Macfarquhar** (1744-1793) and the engraver **Andrew Bell** (1726-1809). The editor was **Wiliam Smellie** (1740-1796) who wrote most of the three volumes with a total of 2391pages. There is no entry of „lightning“, the entry of „thunder“ refers to „electricity“.This entry covers the pages 471 to 485 in Volume II. This indicates that the

connection of both phenomena to electricity was firmly established at that time. On page 480 ten reasons of the similarity between lightning and electricity are listed. Smellie cites mostly the work of the Italian scienti**st Giambatista Beccaria** (1716-1781) but also **Benjamin Franklin** (1706-1790) is often mentioned. Electricity was assumed to be some kind of fluid which ascended out of *subterraneous cavities* into the thunder-cloud. The cloud's shape is correctly described, and it is stated:

*…that thunder-clouds were sometimes in a positive as well as negative state of electricity*.





Cloud-to-cloud lightning is distinguished from cloud-to-ground lightning. Thunder is for the first time explained as collapsing air:

> *One of the principal reasons why those flashes make so long a rumbling, is their being occasioned by the vast length of the vacuum, made by the passage of the electric matter. For though the air collapses the moment after it has passed, and the vibrations (on which the sounds depends) commences at the same moment through the whole length of the track…*


On page 483 under the heading *Method of securing buildings and persons from the effects of lightning* we read the description of a lightning rod:

> *An iron rod placed on the outside of a building, from the highest part continued down into the moist earth, in any direction strait or crooked, following the form of the roof or other parts of the building, will receive the lightning at its upper end, attracting it so as to prevent its striking any other part; and, affording it a good conveyance into the earth, will prevent its damage any part of the building.*


People in a not secured house should avoid sitting near the chimney or window, and sit in the middle of the room on a chair placed on mattrasses or lying on a hammock. - A machine is described on page 476 and plate LXXIV for experiments with

lightning (Fig. 3).

What is missing in this enc. is a reference to **Johann Heinrich Winckler** (1703-1770), Professor in Leipzig, who published a work in 1746 in which he stated that lightning is the same as an electric spark (Winckler, 1746).

Summarizing it can be said that the entries in this enc. make a great difference to former explanations of lightning and thunder, but electricity is not correctly described, because the electron and ions were not known at that time.


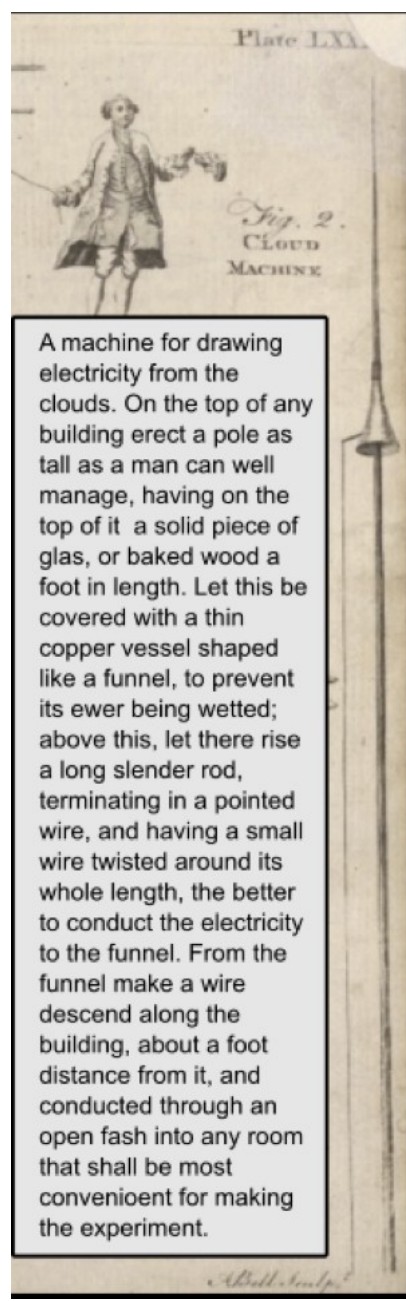

**Fig. 3:** Description of a „machine" for experiments with lightning. Encyclopedia Britannica, 1ˢᵗ edition, extract of plate LXXIV, explanation from page 476. (Enc. Brit. I; 1771).

This was amended in the 11th edition of the ENCYCLOPAEDIA BRITANNICA, issued 1910-11 in 29 volumes
(britannica11, 2024). A great difference to earlier issues was that 1507 authors contributed to the work, edited by the British


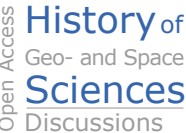

journalist **Hugh Crisholm** (1866-1924). Each entry is signed with the initials of the corresponding author. Our topic is covered under Meteorology and Atmospheric Electricity. The former was written by the British Professor **Charles Chree** (1860-1928), the latter by **Cleveland Abbe** (1838-1916), an American Professor for Meteorology.

The electric processes in the clouds are correctly described as an interaction of particles (Vol. XVIII, p 290), although the
explanation as we know it today is not quite correct:

> *The air — or, properly speaking, the vapour — between cloudy particles —  is generally in a state of supersaturation; ...  the supersaturation must increase steadily until ...  a sudden violent condensation takes place, In which process both the vapour and the cloud particles within a comparatively large sphere are instantaneously gathered into a large drop. The electricity that may be developed in this process may give rise to the lightning*
*flash…*

Reference is given to C.T.R. Wilson (1869-1959), the great pioneer of atmospheric electricity:

> *... and he suggests that the descent of the rain-drops to the ground, carrying negative electricity from the atmosphere to the earth, may perhaps explain the negative charge of the earth and the positive electricity of the atmosphere.*

J.J. Thompson (1856-1940) is cited as well, who earned the Nobel-Prize for the discovery of the electron:

> *I regard electrification of a gas as due to the splitting up of some of the atoms of the gas, resulting in the detachment of a corpuscle from such atoms. The detached corpuscles behave like negative ions, each carrying a constant negative charge which we shall call the unit charge, while the part of the atom left behind behaves like a positive ion with the units positively charged but with a mass that is large compared with that of the negative ion.*

Nothing detailed is said about thunder, except (Vol. XXVI, p 898 ):

> *THUNDER, the noise which accompanies or follows a flash of lightning, due to the disturbance of air by a discharge of electricity.*

Under the heading „Atmospheric electricity" (Vol. II,8, p. 868-870) there are numerous statistics about diurnal and annual frequency of lightning at various places on Earth and about deaths due to lightning.

300                                         - - -

A remarkable excursus to a non-natural science enc. is a brief look at the ENCYCLOPEDIA OF FAIRY TALES, a German enc. on international folkloristic, started in the 1960 as ENZYKLOPÄDIE DES MÄRCHENS (De Gruyter, Berlin 1999, and 2017), and continuously expanded until 2015. It was a project of the Göttingen Academy of Sciences, contains about 3900 articles in 15 volumes  from over 800 authors of over 60 countries.

The entries Thunder and Lightning describe mythological and religious traditions of these natural phenomena from various cultural groups, assigning gods to both items. Folkloristic tales and legends are reported. Several examples are given that GOD intervenes warningly, punitively, and vengefully in earthly matters, in order to avenge violations of divine commands. Different plants and trees which prevent damage by lightning are mentioned, as well as prohibition of certain activities during thunderstorms like cursing, eating, or dancing. - References are given to all details.

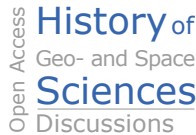


- - -

The entry „Lightning" in the English-language WIKIPEDIA summarizes our present knowledge of this phenomenon. A pdf-representation of the entry covers 36 pages with 165 references. The last addition was made on 1. April 2024.

Particularly the electrification in the cloud is described in more detail. although it is stated that it is still under further
investigation.

*The main charging area in a thunderstorm occurs in the central part of the storm where air is moving upward rapidly (updraft) and temperatures range from −15 to −25 °C . … In that area, the combination of temperature and rapid upward air movement produces a mixture of super-cooled cloud droplets (small water droplets below freezing), small ice crystals, and graupel (soft hail). The updraft carries the super-cooled cloud droplets and very*
*small ice crystals upward.*

*The differences in the movement of the precipitation cause collisions to occur. When the rising ice crystals collide with graupel, the ice crystals become positively charged and the graupel becomes negatively charged,… The updraft carries the positively charged ice crystals upward toward the top of the storm cloud. The larger and denser graupel  is either suspended in the middle of the thunderstorm cloud or falls toward the lower part of the storm.*
*The result is that the upper part of the thunderstorm cloud becomes positively charged while the middle to lower part of the thunderstorm cloud becomes negatively charged.*

Further details are give about the frequency of occurrence (see Fig. 4, as an example)., e.g. :

*On Earth, the lightning frequency is approximately 44 (± 5) times per second, or nearly 1.4 billion flashes per year and the median duration is 0.52 seconds …*

*The place on Earth where lightning occurs most often is over Lake Maracaibo, wherein the Catatumbo lightning phenomenon produces 250 bolts of lightning a day.*

The physical processes within a lightning flash are explained as well, e.g. the leader, providing the  conductive channel for the stroke (although it is stated that initiation of the leader is not well understood so far), upward streamer, finally the return
stroke, the most luminous part of the discharge, followed by several re-strikes. The electric current of a stroke amounts on the average to about 30 kA. The core temperature of the plasma during the return stroke can exceed 27,800°C. Well explained are also the different types of lightning, intra-cloud, cloud-to cloud, and cloud-to-ground, as well as the difference of positive and negative lightning.  It is also stated that the energy of a lightning stroke amounts on the average to between 200 Megajoule to 7 Gigajoule, and the rate at which the return stroke current travels has been found to be around 100,000
km/s (one-third of the speed of light).

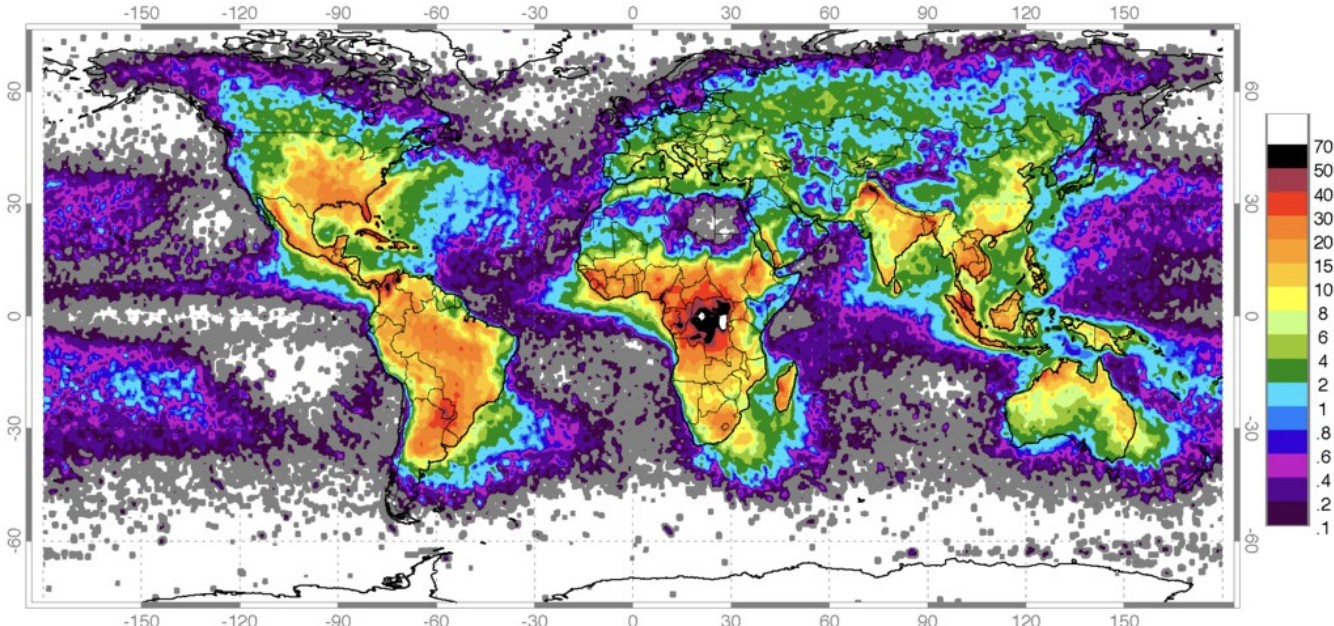

**Fig. 4:** Worldwide lightning strikes, data obtained from April 1995 to February 2003 from NASA's Optical Transient Detector. Public domain, WIKIMEDIA Commons.

What is completely new compared to former enc. is the description of electromagnetic radiation emitted besides visible light:
radio-frequency pulses, x-rays and terrestrial gamma rays.

What is missing in this WIKIPEDIA entry are informations about high speed lightning photography with Boys' streak cameras (Boy, 1926), as well as details and possible explanations of the newly discovered upward lightning strikes into the ionosphere, e.g. elves and blue jets (e.g. Füllekrug et al., 2007), sprites are just mentioned in a half-sentence. - About ball lightning an extra article is available in WIKIPEDIA.


The entry „Blitze" in the German-language WIKIPEDIA is somewhat shorter (21 pdf-pages, 64 references) but treats the newly discovered upwards lightning in more detail. Apart from all the physical details covered in the English-language WIKIPEDIA, it presents lightning statistics for Austria, Germany and Switzerland.

In the French-language WIKIPEDIA the entry „Foudre" has about the same size as the German-language one (20 pdf-pages with 149 references). The content and structure is similar, except for the country statistics.

All three enc. mention various research projects for lightning studies, detection methods, lightning protection and impact on buildings and on biological systems including humans. They all treat religious and cultural aspects of lightning, as well as lightning on other planets.

360                                        ----
Finally, a" special" enc should be briefly mentioned: the 6-volumes ENCYCLOPEDIA OF ATMOSPHERIC SCIENCES (Holton et al. 2003) which is currently updated. The entry Lightning in Vol. 3, authored by M.B. Baker, describes on 17 pages the cloud electrification, details of the flashes, in similar detail as in the WIKIPEDIA, but also lightning climatology.

Additional articles address in Vol. 2, under the heading Atmospheric Electricity the entries: Global Electric Circuit (author E.R. Williams), Ions in the Atmosphere (authors R.G. Harrison & K.I. Aplin), and Sprites (author W.A. Lyons) in total on 23 pages. These topics are described in considerably more detail than in the WIKIPEDIA.

## 6. Concluding remarks

For over two millennia the authors of enc. entries about thunderstorm effects did hardly dispute their ideas. They were regarded as true and correct, because former „authorities" had named them as well. Only from enlightenment onwards authors admit doubts and uncertainties in their texts. In the present WIKIPEDIA, is it stated at several places that there are still many aspects of these phenomena are not yet fully understood. Among those are some details of the cloud electrification and charge separation, the triggering of leaders, the role of infrasound in thunder, details of the terrestrial gamma-ray flashes

and the physics of pearl lightning and ball lightning. Since the WIKIPEDIA entries are continuously refined, it is expected that these uncertainties will be addressed in the future.

**Competing interests:** The author is a member of the editorial board of HGSS.

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
