# Peer review of "Lightning and thunder explanations in encyclopedias from ancient Greece to WIKIPEDIA"

_History of Geo- and Space Sciences, 2024_

## Referee Comment (RC2)

**Reviewing comments**

for manuscript HGSS-2024-9  by Kristian Schlegel

"Lightning and thunder explanations in encyclopedias - from ancient Greece to WIKIPEDIA"

**General impression:**

This reviewer comes from the non-atmospheric-electricity part of the IAMAS/IUGG community and is representative for the manuscript's impression on an informed non-specialist. The paper is regarded as a most interesting collection of state-of-the-art references that appeared over the centuries in encyclopedias regarding the topic "Lightning and thunder". This specialty within the much broader field of atmospheric sciences regularly attracted the attention of the general public. The selection of the material presented in the four main sections appears to be broad, but necessarily subjective.

The main shortcoming of the manuscript about a broad collection of secondary sources and some fine reproductions of old figures is seen in the omission of a convincing motivation for the collection assembled in section 2 to 5 as well as insufficient conclusions in the final section 6.

Ideally, the revised manuscript should mention the author's personal motivation for undertaking his collective exercise (*e.g.*, long personal interest in the topic "lightning and thunder"; a compact specialty of science combined with a long history of publications; encyclopedias as secondary sources ease to selection, etc.). Likewise "lessons learnt" should be dealt with in some detail in the final section and possibly also touch upon the human enterprise "scientific study" (every generation mostly pretends to have reached sufficient understanding, even if often overwhelmed by the complexity of [atmospheric] nature).

I would like to leave it to the topical editor to advise the author with guiding hints to be followed during the production of a revised version of the manuscript.

**Specific observations from reading the manuscript:**

*Section 1: Introduction (lines 17–44):*

Here mainly technical details about the considered encyclopedias are collected. Missing are the personal motivation for the collection and the expression of any hope about the general usefulness of the collection. Can the combination of the topic "Lightning & thunder" with enc.s as solitary source of information be expected the provide general insights about the acquisition of knowledge? Are similar attempts for other topics known to exist? If yes, examples should be mentioned.

At least an additional half to full page would assist the reader a lot.

*Section 5: Enlightenment and later (lines 199–366):*

It appears to be appropriate to split this section into two:
*5. (Early) Enlightenment (1700 – 1900)* and
*6. Modern times (after 1900?)*
with thresholds adjusted as appears to be most fitting.

At the end of (new) section 6 reference to a 12-year old research article (Schmidt et al. 2012) could be made as it attempted to closely link lightning in a (isolated) thunderstorm to its life-cycle and the complexity to atmospheric dynamics. Using Fig. 4 (of Schmidt) as a local complement to the global view given in the current Fig. 4 would provide the reader with a flavor of the breath of scales that are dealt with in current "lightning research". The case study occurred over the Black-Forest region and was published in another journal of the Copernicus family of journals.

*(Current) Section 6:    → 7. Concluding remarks (lines 369–376):*

As stated above (for *Introduction*) some more specific conclusion should be assembled. They may comprise:
- How can the route of progress be classified? Purely random? Parallel to the progress in classical physics? How important is a sufficiently synoptic view around the dynamic generation of thunderstorms?
- How useful proved to be the restriction to encyclopedias as sole sources (except the recent example of a research article)?
- Are the general comments about the human endeavour of collecting the entire body of current knowledge within printed multi-volume encyclopedias (from ~1700) to the multi-nation, multi-authors exercise of Wikiedia (since 2003)? Just useful or also containing some human hubris?
- A mention of the scientific grouping ICAE (International Commission on Atmospheric Electricity; https://www.iamas.org/icae/ ) within IAMAS as a current home for the topic the evolution of which is presented.in the article.

**Reference:**

Schmidt, K. et al., 2012: Detailed flow, hydrometeor and lightning characteristics of an isolated thunderstorm during COPS. Atmos. Chem. Phys., **12**, 6679–6698; online: www.atmos-chem-phys.net/12/6679/2012/ , doi:10.5194/acp-12-6679-2012

---

## Author Comment (AC1)

I am thankful to Ref.1 for his valuable criticism and his suggestions to improve my manuscript. My point-by-point answers follow:

*This is an interesting perspective on encyclopedias from the perspective of the description of lightning and thunder. It contains a range of worthwhile material, and it should be published in HGSS.*

*I have two principal points to make about it. Firstly, the motivation for the work is not clearly enough provided at the outset. For example, the material at L38-41 could be given in the opening sentences of section 1. Secondly, the structure of the article needs to be outlined early on, probably in section 1.*

I added the following sentences at the beginning of Chapter 1:

The motivation of this work was to study the development of descriptions and explanations of lightning and thunder over the past two and a half millennia. A convenient source for this purpose are encyclopedias, since they are supposed to contain the actual knowledge of the respective age which has been accepted by the authorities of the time, relaying often to earlier scholars. Their widespread circulation indicates that the readers appreciated exactly this purpose.

*Currently, the sections are chronological, which is fine. But a guide to what is coming - perhaps even a simple table - would help the reader further.*

At the end of Chapter 1 I added:

 Starting with the eminent Greek philosopher Aristotle in Chapter 2, we examine a few Roman enc. and several medieval enc. in Chapters 3 and 4. Chapter 5 deals with enc. of enlightenment and following times, and finally Chapter 6 with online- and a special enc.

*L9 (and L12) "wrong" implies hindsight evaluation - better avoided. Just acknowledge that the medieval view was different to the modern view, and perhaps where it originated.*

in L9 I changed „wrong" into quite different from the modern view and in L12 „wrong and strange" into from our present understanding unorthodox.

*L16 Are all details of physical phenomena ever completely understood? It seems unfair to thunderstorm scientist to single them out.*

I changed the sentence to Finally, it is stated that even today several details of thunderstorms are not well understood

*End of Sect 1. Please add a few lines explaining the structure of the paper and how the subsequent sections are organised.*

see above

*L60 Just point out that, at the time, they considered thunder to precede lightning. (The early authors would have had their own reasons, and it is only erroneous by modern investigations).*

I changed the sentence to Here the most important assumption is that thunder precedes lightning, contrary to the modern view.

*L95 Only some observations of ball lightning describe its ability to move through walls, so give a reference for this. A useful authoratative reference is Pippard 1982 https://www.nature.com/articles/298702b0*

I included the reference and changed the sentence to … could be a ball lightning, since some observations describe its ability to move through walls (e.g. Pippard 1982):

*L268 Electricity was described in the terms of the time: a modern judgement on its validity, requiring ions and electrons, is inappropriate here.*

I changed the sentence to …to former explanations of lightning and thunder, although our modern view of electricity is different.

*L277 Chree was superintendent of Kew Observatory (see MacDonald, 2018) and therefore a knowledgeable establishment figure. As was Abbe. Having such contributors may have caused a transformative step in quality.*

I added here … British Professor and superintendent of the famous Kew Observatory **Charles Chree**

*L300-309. This seems to be a footnote, or possibly, an endnote. Please be clear about how it links to the other material.*

I cannot follow this suggestion because I think that mentioning the enc. of Fairy Tales is a remarkable supplement to this work. To make this clear I started this section with: In order to supplement the so far treated enc with a completely different, non-natural science perspective of our phenomena, a brief excursus to the ENCYCLOPEDIA OF FAIRY TALES is included. It is a German enc…

*L310. This needs to be identified as a separate section, for Online Encyclopedias. There is a point to be made about distributed authorship, in comparison with the authorships of the earlier material. Distributed authorship brings other issues - sometimes Wikipedia articles are poor and dominated by the erroneous views of one or a few non-expert individuals.*

I started a new section here with:  6. Online Encyclopedias and a „Special Encyclopedia"

It should be emphasized that online enc. have a distributed authorship, quite different from the enc. treated so far where the authors are known. This means that WIKIPEDIA articles can contain questionable views of individuals.

*Minor points*

*L23 Encyclopedia*

*L31 angel*

*L152 Worth mentioning…*

*L216 speed (not distance)*

*L237 …of environmental physics*

*L347 Boys (The originator's name was Charles Vernon Boys).*

**All corrected**

---

## Author Comment (AC2)

I thank the referee for his valuable comments to improve my manuscript.

Here is my reply:

*Section 1: Introduction*

The other referee asked for a „personal motivation" as well. I therefore included the following paragraph at the beginning of the introduction.

The motivation of this work was twofold. Fascinated by the ancient and modern encyclopedias as a summary of contemporary knowledge and continuing interest in all phenomena related to lightning and thunder, it suggests itself to study the development of descriptions and explanations of lightning and thunder over the past two and a half millennia in encyclopedias. They are supposed to contain the actual knowledge of the respective age which has been accepted by the authorities of the time, relaying often to earlier scholars. Their widespread circulation indicates that the readers appreciated exactly this purpose.

*Section 5: Enlightenment and later*

To my knowledge an „end" of enlightenment is not clearly defined. Therefore I combined the encyclopedias dealt within in this section.

*Current Section 6: Concluding remarks.*

I rephrased this section as follows:

It should be emphasized that the explanations of lightning and thunder in enc. is certainly not the only and the most appropriate way to document the progress in this field. But it was an interesting endeavor regarding the author's interest in enc. For over two millennia the authors of enc. entries about thunderstorm effects did hardly dispute their ideas. They were regarded as true and correct, because former „authorities" had stated them as well. This could be regarded as a kind of human hubris. Only from enlightenment onwards authors admit doubts and uncertainties in their texts. In the present WIKIPEDIA, is it stated at several places that there are still many aspects of these phenomena are not yet fully understood. Among those are some details of the cloud electrification and charge separation, the triggering of leaders, the role of infrasound in thunder, details of the terrestrial gamma-ray flashes and the physics of pearl lightning and ball lightning. Since the WIKIPEDIA entries are continuously refined, it is expected that these uncertainties will be addressed in the future.

There is no doubt however, that important progress in the understanding of lightning, thunder and related phenomena is published beyond enc. In particular, studies employing modern tools like Doppler radar, lightning detection by emitted radio waves and multichannel spectral measurements from satellites can greatly enhance our knowledge of the complicated details of thunder and lightning. A remarkable example of such a study was published by Schmidt et al (2012). Studies initiated and conducted within ICAE (International Commission on Atmospheric Electricity) a sub-group of IAMAS (International Association of Meteorology and Atmospheric Sciences) are expected to follow.